# Combination of Optical Biopsy with Patient Data for Improvement of Skin Tumor Identification

**DOI:** 10.3390/diagnostics12102503

**Published:** 2022-10-15

**Authors:** Yulia Khristoforova, Ivan Bratchenko, Lyudmila Bratchenko, Alexander Moryatov, Sergey Kozlov, Oleg Kaganov, Valery Zakharov

**Affiliations:** 1Laser and Biotechnical Systems Department, Samara National Research University, 34 Moskovskoe Shosse, 443086 Samara, Russia; 2Department of Oncology, Samara State Medical University, 89 Chapaevskaya Str., 443099 Samara, Russia

**Keywords:** Raman spectroscopy, cancer risk factors, skin cancer, PLS analysis, statistical significance

## Abstract

In this study, patient data were combined with Raman and autofluorescence spectral parameters for more accurate identification of skin tumors. The spectral and patient data of skin tumors were classified by projection on latent structures and discriminant analysis. The importance of patient risk factors was determined using statistical improvement of ROC AUCs when spectral parameters were combined with risk factors. Gender, age and tumor localization were found significant for classification of malignant versus benign neoplasms, resulting in improvement of ROC AUCs from 0.610 to 0.818 (*p* < 0.05). To distinguish melanoma versus pigmented skin tumors, the same factors significantly improved ROC AUCs from 0.709 to 0.810 (*p* < 0.05) when analyzed together according to the spectral data, but insignificantly (*p* > 0.05) when analyzed individually. For classification of melanoma versus seborrheic keratosis, no statistical improvement of ROC AUC was observed when the patient data were added to the spectral data. In all three classification models, additional risk factors such as occupational hazards, family history, sun exposure, size, and personal history did not statistically improve the ROC AUCs. In summary, combined analysis of spectral and patient data can be significant for certain diagnostic tasks: patient data demonstrated the distribution of skin tumor incidence in different demographic groups, whereas tumors within each group were distinguished using the spectral differences.

## 1. Introduction

The annually growing trend of melanoma disease is observed worldwide [1]. Research [2] estimated that 106,110 new cases of melanoma were diagnosed and about 7180 people died of this disease in the USA in 2021. The growth of melanoma can be caused by different personal [3,4,5], behavioral, and socioeconomic factors [6,7]. The National Cancer Institute has reported [2,8,9,10] that melanoma is more common in men than women and more frequent among whites in comparison with other races or ethnicities. Moreover, there is a strong relationship between melanoma cases and patient age [2]. For example, incidence rates for MM skin cancer in the UK are the highest in people aged 75 and over [11].

In terms of environmental factors, ultraviolet radiation is the most dangerous factor causing melanoma growth [11,12]. Localization can also be a potentially informative factor for more accurate skin cancer diagnosis, because some types of skin tumor often develop in the body areas that are directly exposed to UV radiation, with others appearing in covered body sites subjected to intense sunburn because of their rare exposure to regular UV radiation [13].

High risk can also be associated with family history: about 10% patients with melanoma have a family history of the disease [14,15,16,17]. The study by Hemminki et al. [18] demonstrated that melanoma is several times more common in people whose first-degree relatives have had melanoma. Moreover, researchers [19,20] reported on the relationship between patient history of skin neoplasm and the risk of developing melanoma, suggesting that the patient’s history indicates risk of skin cancer growth. People working in certain professions can have a higher risk of skin cancer and some precancer conditions, due to interaction with dangerous industrial carcinogens [21].

Preliminary diagnosis of melanoma using dermoscopy [22] or other developing optical biopsy techniques [23,24,25] did not consider the above risk factors that may be potential prerequisites for developing skin cancer. However, incorporating patient-specific information can improve the accuracy of disease identification based on clinical studies [26,27,28]. Pacheco and Krohling [26] demonstrated the importance of clinical features for skin cancer detection based on clinical images and confirmed the hypothesis that patient clinical information is important for this task. However, they concluded that the clinical features they examined were not practical indicators for all types of skin lesions. Zeng et al. [27] examined skin tumors using Raman spectral data, considering various risk factors, and revealed that only patient age significantly contributed to improved diagnosis of malignant tumors. Taking into account the findings of other research teams [26,27,28], we aimed to test the possibility of improving skin cancer identification with our experimental data, by combining Raman and autofluorescence data as well as patient information.

In our previous work [29,30], we performed an optical biopsy using Raman and autofluorescence (AF) spectroscopy to diagnose skin cancer. Raman spectroscopy has proved to be a sensitive research instrument in clinical practice for a number of purposes [24,27,30,31]. The proposed method [29,32] was able to classify skin neoplasms with a mean accuracy higher than the accuracy of general practitioners or trainees, and with comparable or less accuracy than trained dermatologists and experts. Therefore, it remains necessary to improve the accuracy of skin cancer diagnosis performed with Raman and AF analysis.

The aim of our study was to estimate the prognostic possibility of combining individual patient factors with the results of optical biopsy for detecting skin cancer. The spectral data of 617 skin tumors that were analyzed in our previous work [29] were combined with data on risk factors, for joint analysis. We demonstrated the results of the proposed approach by combining Raman spectroscopy and AF with individual patient factors such as environmental risk, history, and personal risk factors for classification of malignant skin tumors, melanoma, and other skin neoplasms.

## 2. Materials and Methods

### 2.1. Experimental Setup

The detailed description of the experimental setup for simultaneous Raman and AF signal registration was presented in our previous studies [29,32]. The scattering spectral response from skin tissue in the near-infrared region was stimulated using a thermally stabilized diode laser module (LuxxMaster, LML-785.0RB-04, PD-LD, Ushio Inc., Tokyo, Japan) with 785 ± 0.1 nm central wavelength. The laser power density on the skin was about 0.3 W/cm^2^ and did not cause any damage to skin or discomfort in patients. The optical Raman probe (RPB785, InPhotonics Inc., Norwood, MA, USA) contained supplying and collecting branches. Laser excitation at 785 nm was delivered to the skin surface by means of the excitation optic fiber (0.22 NA, 100 μm) and the supplying branch of the probe with a band-pass filter and a focusing lens. The scattered radiation was collected by the same lens and delivered to the collecting branch by the dichroic mirror and the conventional mirror. The longpass filter cut the excitation laser wavelength from the collected signal, and the Raman and fluorescence signals of skin tissue were transmitted to the spectrometer using the focusing lens and the collecting fiber (NA 0.22, 200 μm). The collected signal was decomposed into a spectrum using a portable spectrometer (QE65Pro, Ocean Optics Inc., Largo, FL, USA). The spectra were registered in the 780–1000 nm region with spectral resolution of 0.2 nm. The acquisition time was 20 s with a triple accumulation. The QE65Pro detector was cooled down to −15 °C. The silicon tip on the probe provided the 7–8 mm distance between the skin surface and the probe for all measurements.

### 2.2. Patients

The protocols of the in vivo tissue diagnostics were approved by the ethical committee of Samara State Medical University (Samara Region, Samara, Russia, protocol No 132, 29 May 2013), the clinical studies fall within The Code of Ethics of a Doctor of Russia, approved at the 4th Conference of the Russian Medical Association, and within the World Medical Association Declaration of Helsinki. The study involved 615 patients of different ages, including 178 men and 437 women, who consulted specialized oncologists in the Samara Regional Clinical Oncology Dispensary from May 2017 to December 2019. All the patients were aged ≥ 18. Informed consent was acquired from all patients before the in vivo study.

Spectral measurements of 617 tumors were carried out for 615 patients. The spectral measurement of each skin tumor was registered from the approximate central point of the tumor area. The region of interest for spectral registration of tumors was confirmed by a medical specialist on the basis of dermatoscopic images. The skin tumors were localized at different body sites. The sizes of skin tumors varied widely, from 0.3 to 5 cm. Summary of the patients and tumors is presented in [29]. In accordance with results of histopathological analysis, the analyzed spectral cohort included 204 malignant tumors (70 malignant melanomas (MM), 122 basal cell carcinomas (BCC) and 12 squamous cell carcinomas (SCC)), as well as 413 benign tumors (26 dermatofibromas (DF), 62 papillomas (PP), 40 hemangiomas (HE), 113 seborrhoeic keratosis (SK), 170 nevi (NE) (all types), 1 cutaneous horn, and 1 benign tumor of epidermal appendage).

### 2.3. Risk Factors for Skin Cancer Growth

Cancer develops when human cells are damaged due to various factors and the number of damaged cells starts to grow uncontrollably. In this work, we analyzed several risk factors that can potentially provoke skin cancer growth.

At the initial appointment, the oncologist collected the patient history and potential risk factors for skin cancer growth: gender (G), age (A), tumor localization (L), family history (FH), personal history (PH), sun exposure (SE), size (S), and occupational hazards (OH). All the collected demographic indicators were defined by the patient survey. However, for different reasons, not every patient provided the full set of collected risk factors. Only gender, age and localization factors were received for all the 617 skin neoplasm spectra. Therefore, we considered two spectra datasets:
(I)Spectral data of all 617 skin neoplasms with only three indicators: (G), (A), (L);(II)Spectral data of only 481 out of the 617 skin neoplasms with all eight indicators: (G), (A), (L), (FH), (PH), (SE), (S), (OH).

All the risk factors were digitized:

G:1—male; 2—female;A:1—under 29, 2—30 to 39, 3—from 40 to 49, 4—from 50 to 59, 5—from 60 to 69, 6—over 70;L:1—head and neck, 2—trunk, 3—upper limb, 4—lower limb;FH:0—no malignant diseases in close relatives; 1—close relatives with malignant diseases, 2—close relatives with skin cancer disease;PH:0—the patient had no serious disease; 1—the patient had a different disease; 2—the patient had a malignant disease;SE:0—the patient avoids suntan; 1—the patient gets suntan without sunburn; 2—the patient often has sunburn;S:1—from 0 to 5 mm; 2—from 6 to 20 mm; 3—21 mm;OH:0—no occupational hazards; 1—occupational hazards due to skin contact with chemicals (e.g., work with petroleum products, on chemical plants, etc.).

The digitization of the patient factors was performed by specialized oncologists at the Samara Regional Clinical Oncology Dispensary.

### 2.4. Preprocessing and Statistical Analysis of Spectra

The spectra were recorded in the 780–1000 nm region, but only the 803–914 nm spectral region corresponding to the 300–1800 cm^−1^ wavenumber region in terms of Raman spectroscopy was analyzed. Firstly, the raw spectra in the region of interest (803–914 nm) were preprocessed by the following process: smoothing by the Savitsky–Golay filter, normalization by the standard normal variate method (SNV), and centering.

In accordance with the data described in Section 2.2, we considered six classification models with different sets of risk factors:

I.1Malignant (n = 204) vs. benign (n = 413) neoplasms with 3 risk factors;II.1Malignant (n = 157) vs. benign (n = 324) neoplasms with 8 risk factors;I.2MM (n = 70) vs. benign pigmented (Ne and SK, n=283) neoplasms with 3 risk factors;II.2MM (n = 49) vs. benign pigmented (Ne and SK, n = 221) neoplasms with 8 risk factors;I.3MM (n = 70) vs. SK (n = 113) with 3 risk factors;II.3MM (n = 49) vs. SK (n = 90) with 8 risk factors.

Each spectrum included the Raman and AF signals in the region of interest (of 803–914 nm) and, therefore, represented a discrete set of intensity values at the 515 wavelengths (in accordance with the spectral resolution of the spectrometer). For the subsequent regression analysis, the 515 spectral parameters respectively representing each tumor after preprocessing were combined with the corresponding risk factor parameters. Therefore, in classification models (I.1), (I.2), and (I.3) each tumor was represented as 518 predictors (515 spectral parameters and three risk factor parameters) for PLS analysis, and in classification models (II.1), (II.2), and (II.3) as 523 predictors (515 spectral parameters and eight risk factor parameters), respectively.

The experimental data were processed using partial least square discriminant analysis (PLS-DA) [33]. The PLS-DA method was applied to build a regression model between the analyzed tumor predictors and tumor types. Stability of the PLS-DA classification was checked by means of 10-fold cross-validation. The number of latent variables (LVs) for the PLS-DA models was chosen according to the minimum of the RMSE in the 10-fold cross-validation. To estimate the importance of all tumor predictors in the model, variable importance in projection (VIP) analysis was performed [34]. The VIP scores highlighted the informative predictors of tumors in the regression model that were more important for classifying different tumor types. Higher relative intensity of VIP score indicated that the predicted variable was more significant. To determine the differentiation accuracy of the tumor analysis, the PLS predictors were calculated as numeric values of tumor diagnosis in the built regression model.

The results of the skin tumor differentiation were visualized using a bee-swarm diagram and the receiver operating characteristic (ROC) curves plotted using R studio software [35]. The ROC analysis shows the diagnostic performances of the regression model. For quantitative analysis, the area under the curve (AUC) was calculated. The significance of the AUCs and the comparisons between different AUCs were tested in a standard manner [36].

## 3. Results

### 3.1. Malignant vs. Benign Neoplasms

(I.1) To discriminate the malignant (n = 204) vs. benign (n = 413) neoplasms from set (I), the 0.600 (0.567–0.652) ROC AUC was obtained using only the spectral data (RS and AF data). The complementation of spectral dataset (I) with three risk factors made it possible to improve the ROC AUCs to 0.818 (0.778–0.841). Moreover, adding each patient factor separately to the spectral data significantly increased the ROC AUC (see Table 1). The distribution of VIP scores as a weighted sum of loadings is shown in Figure 1, highlighting all spectral features for all loadings obtained in this PLS classification model. For this model, the VIP scores were utilized to classify malignant versus benign tumors by determination of informative predictors (gender (G), age (A), location (L), and 515 spectral parameters) in regression specification. The VIP scores presented in Figure 1 demonstrate that age (A) is the most informative risk factor, which was proved by the most significant improvement of ROC AUC (0.804, *p* = 9 × 10^−9^) when only age was incorporated into the spectral data, compared with the other factors.

The ROC AUCs and the bee-swarm diagram for this classification are presented in Table 1 and Figure 2a–c.

(II.1) For set (II), the classification of malignant (n = 157) vs. benign (n = 324) neoplasms using the PLS analysis was performed with the 0.610 (0.556–0.663) ROC AUC on the basis of only the spectral data, and with the 0.789 (0.746–0.832) ROC AUC when supplying the spectral data with eight risk factors. For this set, age was also the most important risk factor. The ROC AUCs and bee-swarm diagram are presented in Table 1 and Figure 2d–f.

Table 1 presents the ROC AUCs of the models built using all risk factors separately. Improvement of the ROC AUC by incorporating the spectral data with all risk factors to identify malignant skin cancer was statistically significant (*p* < 0.05) in models I.1 and II.2.

### 3.2. MM vs. Benign Pigmented Neoplasms (Ne and SK)

(I.2) In this classification task, regression analysis of the cases from dataset (I) using only the spectral data was performed with 0.690 (0.630–0.761) ROC AUC. For this task, the combined analysis of the spectral data and the three risk factors significantly improved the diagnostic performance to 0.825 (0.766–0.884) ROC AUC. The contribution of all three risk factors in this model was significant for MM identification (*p* < 0.05), whereas separately adding age, gender, or localization did not result in significant improvement of the ROC AUC. Figure 2g–i and Table 1 show the results from cohort (I) for this classification task.

(II.2) In the same classification task for cohort (II), MMs (n = 49) were differentiated from benign pigmented neoplasms (n = 221) with 0.789 (0.718–0.861) ROC AUC using only the Raman and AF spectral data, and 0.849 (0.785–0.914) ROC AUC when combining the spectral and risk factor variables. However, in this case, there were no significant differences (*p* = 0.14) between the ROC AUCs obtained for the models with the eight risk factors and those without. Figure 2 presents only the statistically significant results and therefore does not include a diagram for this model. Table 1 indicates the ROC AUCs for this classification task.

### 3.3. MM vs. SK

(I.3) When classifying the MMs (n = 70) vs. SKs (n = 113) from cohort (I) using the spectral data, 0.791 (0.722–0.859) ROC AUC was obtained. When the three risk factors were combined with the Raman and AF spectral data, the ROC AUC increased to 0.844 (0.786–0.902) but with no statistical significance.

(II.3) When analyzing dataset (II) with the same classification task, the discrimination model showed 0.820 (0.748–0.892) ROC AUC obtained by combining spectral data and the eight risk factors, while the analysis of only the spectral data was performed with 0.814 (0.740–0.888) ROC AUC.

The AUCs of this classification task were insignificantly improved (*p* > 0.05) in models I.3 and II.3. The diagnostic performance results for this classification task are presented in Table 1.

## 4. Discussion

The classification results for different types of skin neoplasms based on Raman and AF spectral data were demonstrated in our previous research [29,32]. In this current work, we combined individual patient risk factors and spectral data to obtain a more precise skin cancer diagnosis, in particular of malignant tumor and MM. It should be noted that true statistics of skin cancer incidence might differ from the data we obtained, due to including in this study only those patients who were aware of skin tumors, attentive to their health, and had access to resources for tumor detection.

Considering the significance of the analyzed patient factors, our classification models have demonstrated that although gender was a significant factor for classifying malignant versus benign skin tumors in both model I.1 and model II.1, it was not significant for diagnosis of MM. The analyzed cohort was heterogeneous in the numbers of men and women: women outnumbered men 2–3 times in the general cohort and in the analyzed classes. However, the proportion of men with malignant tumors among all men involved in this study was 0.43, whereas the relative proportion of women was 0.28. At the same time, the relative incidence rates of MM were 0.13 among men and 0.11 among women (Appendix A). The statistics for different ethnicities and races vary. According to statistics from Australia and the USA [10,13], the incidence of MM is higher among men than women. In Russia in 2020, the standardized incidence rates (number per 100,000 population) of skin cancer (without MM) and MM among men were 21.48 and 4.08 respectively, whereas among women the figures were 20.62 (skin cancer without MM) and 4.32 (MM) [37]. Data for our cohort was collected from May 2017 to December 2019, and revealed that in our study the relative number of malignant cases was higher among men, while the number of MM cases was the same among men and women.

Localization was also a significant factor in models I.1 and II.1 for malignant versus benign classification. We suppose that the significance of this factor can be explained by the most common localization group for different tumor types (Appendix A). In model I.1, most of the malignant tumors, namely 86 out of 204 cases (about 42%), were located on the head and neck, while 209 out of 413 benign tumors (about 51%) were situated on the trunk. Despite the fact that more MM cases occurred on the trunk (about 51%), the large number of malignant tumors on the head and the neck was due to the contribution of the BCC and SCC cases. However, when classifying MM and benign pigmented tumors or SK, this factor was found to be insignificant, because most cases within each class occurred on the trunk: 51% among melanoma cases, 51% among benign pigmented cases, and 47% among cases of seborrheic keratosis.

The sun exposure factor was insignificant in all models (II.1, II.2, and II.3), but exposure to sun radiation is partly responsible for localization. BCC and SCC are more likely to occur on body areas that are most exposed to solar radiation, i.e., on the head. According to our data presented in Appendix A, 61% of BCC and 71% of SCC in this study were localized on the head and the neck [29]. On the other hand, most MM (51%) and other melanocytic tumors, such as pigmented nevus, occurred on the trunk and legs: these body areas may be subjected to intense sunburn because of less frequent exposure to regular UV radiation. Other research [38] reports that trunk melanomas are more strongly associated with pigmented nevus counts. Thus, exposure to sun radiation is an equally important growth factor for all melanocytic tumors, confirmed by a similar distribution of various melanocytic tumors (e.g., most cases of MM and benign pigmented nevus were recorded on the trunk). Therefore, these tumors were not distinguished within each localization.

Our models suggest that age is a significant factor when classifyin malignant and benign tumors, because the patient distribution in each age group was different. Most cases of malignant tumors (about 44%) were recorded in patients over 70, while the groups of patients with benign tumors aged from 30 to 39, from 40 to 49, from 50 to 59, and from 60 to 69 were equal in size. For classification of MM versus only SK, age was completely insignificant and did not improve the ROC AUC when added to the spectral data, because the distribution of patients in age groups for MM and SK was fairly similar (Appendix A). The maximum frequency of MM and benign pigmented tumors (Ne and SK) was recorded in the age group from 60 to 69. The larger number of pigmented benign tumors in the age group from 60 to 69 was due to more SK cases, which did not allow significant separation of these classes by age. However, there were differences in the numbers of patients in other age groups: among those with benign pigmented tumors, more than 30% were under 40 because of a larger number of young patients with pigmented nevus. This resulted in the fact that adding the age factor to the spectral data improved the ROC AUC to 0.808 (0.734–0.881) for the classification of MM versus benign pigmented tumors, but these differences were not sufficient for statistical significance.

Finally, it should be noted that in all three classification tasks (models II.1, II.2, and II.3), OH, FH, and PH were not able to improve the ROC AUCs (see Table 1). We lack precise data on the behavioral and genetic factors obtained in the survey because the oncologists collected this information by questioning the patients, which can lead to inaccuracy and uncertainty. To enhance the importance of these factors, they could be defined in a more precise way, for example, in terms of controlling exposure to sunlight or to chemicals that can be dangerous in case of contact with skin. Thereby, the significance of such patient data as gender, age, tumor localization, and size can become more reliable for tumor detection.

Diagnostic performance combining patients’ demographic data with optical data has been evaluated in several works [26,27,28]. Zhao et al. [27] investigated whether incorporating such patient demographics as gender, skin type, localization, and age into Raman spectral analysis can improve performance in malignant skin cancer diagnosis. Using PLS analysis, the authors reported that the ROC AUC improved significantly from 0.913 (0.892–0.933) to 0.934 (0.917–0.952) (*p* < 0.05) after combining only the Raman data with all the demographics, to differentiate malignant and benign skin lesions. In comparison with the study by Zeng et al., in our work we analyzed a larger set of risk factors for cancer growth, including not only demographics but patient lifestyle and behavioral factors as well. We similarly found that combining all the risk factors with the spectral data achieved better performance in discriminating malignant and benign tumors, increasing the ROC AUC from 0.600 to 0.818 with three factors and from 0.610 to 0.789 with eight factors. We increased the ROC AUCs by 30–36% taking into account the patient risk factors, while in the study by Zeng et al. [27] the improvement was only 2%. Probably, this greater improvement of malignant skin cancer identification was a result of a different signal-to-noise ratio in the spectral data. Our spectral data were recorded with a lower signal-to-noise ratio [39] that resulted in low accuracy of detecting malignant neoplasms by only the Raman and AF spectra (0.600 and 0.610 ROC AUC in models I.1 and II.1, respectively), and a significant improvement when the patient factors were added. In the previous work [27], the authors used a highly sensitive spectroscopic system that allowed them to obtain a high ROC AUC 0.913 using only the Raman spectral data.

Kharazmi et al. [28] proposed a non-invasive fast BCC detection tool that incorporates dermoscopic lesion features and clinical patient information including lesion localization, size, and elevation, as well as patient age and gender. The integrated analysis of the patient profile and dermoscopic features using data-driven feature learning allowed them to increase the ROC AUC for BCC detection from 0.847 to 0.911, in comparison with only the dermoscopic features. According to our statistics [29], BCC cases occur within specific demographic conditions, for example, 61% of BCCs were located on the head and neck, and 90% of BCCs were recorded from patients over 60. Thus, it might be assumed that we would be able to determine the significance of patient factors for BCC detection. However, we analyzed BCC and MM as malignant tumors and did not estimate the importance of patient information for identifying BCC only. Considering that our statistical results about BCC are in good agreement with the statistics reported by Kharazmi et al., this may suggest the significance of patient factors only for several types of skin lesions.

It is also interesting to compare the results of the proposed methodology and the results of dermoscopic image analysis performed by dermatologists, which represents the current standard for clinical diagnosis of skin lesions. According to research [40], the accuracy of melanoma vs. non-melanoma skin lesion classification was 79.9% for novice dermatologists, 83.3% for qualified dermatologists, and 86.9% for experts. The mean diagnostics performance of 21 board-certified dermatologists using dermoscopic images to classify 71 malignant vs. 40 benign lesions was nearly 71% sensitivity and 81% specificity [41]. Thus, the proposed methodology can classify skin neoplasms with a mean accuracy higher than GPs and trainees, but with slightly less accuracy than trained dermatologists and experts.

To sum up, our results show that information on patient risk factors and Raman and AF spectral data can complement each other to provide more accurate skin cancer identification. For each skin tumor type, we observed a specific distribution trend by gender, age group, and localization, in good agreement with worldwide statistics on skin tumor incidence. Patients’ age and tumor localization are able to discriminate tumors in different groups, but these factors become insignificant when analyzing different skin tumors within individual groups. For example, similar numbers of malignant, benign, pigmented tumors, SK, and MM were recorded on the trunk or in patients aged from 50 to 59 or from 60 to 69. So, within a separate demographic group, accuracy results for different tumor type diagnosis can differ when only the Raman and AF spectral data are used. For this reason, to differentiate malignant versus benign skin tumors we improved the ROC AUCs by adding risk factors to the model. To differentiate MM versus pigmented skin tumors or SK, similar demographic trends did not allow us to increase the performance accuracy of skin tumor identification. To improve diagnostic performance, the proposed methodology may be added to the estimation of neoplasm morphology performed during dermoscopy analysis. Deep learning-based applications using computer visualization have shown promising results in detecting melanoma based on the analysis of dermoscopic images [40,41,42]. However, additional studies are required to estimate the capability of joint dermoscopy analysis and low-cost Raman systems.

## 5. Conclusions

We tested the possibility of improving skin cancer detection by combining spectral analysis with analysis of individual patient characteristics and factors for skin cancer growth. We analyzed two cohorts of patients with skin tumors: (I) the cohort with 617 spectra of different tumors and three patient factors for each case, and (II) the cohort with 481 spectra of different tumors and eight risk factors. For each cohort, three classification tasks were considered: malignant versus benign tumors, MM versus benign pigmented tumors, and MM versus SK.

The significance of risk factors for type of cancer growth was estimated when all factors were combined with the spectral data, and when each factor was added separately to the Raman and AF spectral data. Statistical improvement was achieved for the classification of 204 malignant tumors and 413 benign tumors, from 0.610 to 0.818 ROC AUC, *p* = 2 × 10^−11^, when spectral data in the 300–1800 cm^−1^ range were combined with three individual patient factors for skin cancer growth. Moreover, classification of 157 malignant tumors and 324 benign tumors using the spectral data and eight risk factors was statistically improved from 0.610 to 0.789, *p* = 5 × 10^−7^. Finally, 70 MMs and 283 benign pigmented skin neoplasms were differentiated with a statistical improvement from 0.709 to 0.825, *p* = 0.02 when combining the spectral data and the three risk factors. Improvements of ROC AUC for discriminating MM (n = 49) and pigmented benign tumors (n = 172) with eight factors, MM (n = 70) and SK (n = 113) with three factors, and the MM (n = 49) and SK (n = 90) with eight factors were all statistically insignificant.

Our results show that among all risk factors, patient demographics including gender, age, and tumor localization were statistically significant for detecting skin tumor type, due to their univocal definition. In contrast, the data for behavioral factors were collected by staff directly from patients and might therefore lack accuracy. For certain classification tasks, it was found that the combination of spectral data and patient risk factors was significant. Particular overall trends for each skin tumor type were observed for patient age, gender, and tumor localization. However, these demographic features did not allow us to discriminate different tumor types, especially pigmented tumors, within an individual demographic group. Therefore, distinguishing skin tumors in groups with similar demographics was possible using the Raman and AF spectral data only. However, these findings need to be verified in further experimental cohort studies.

## Figures and Tables

**Figure 1 diagnostics-12-02503-f001:**
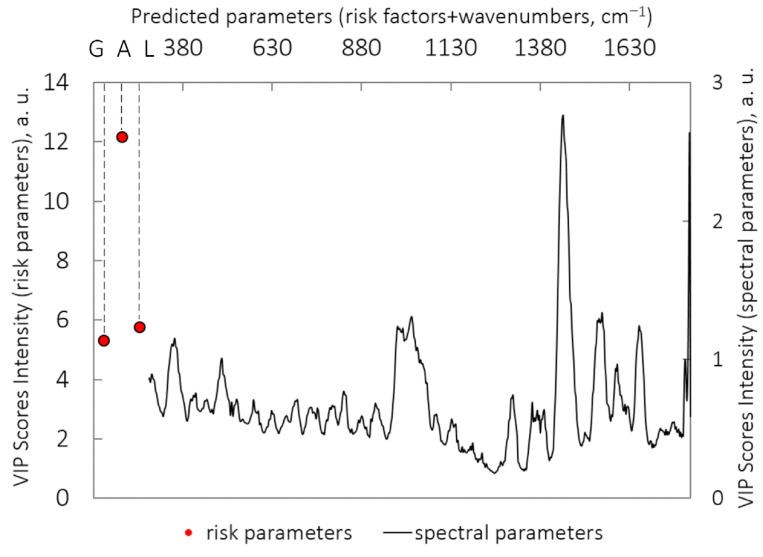
VIP scores for PLS-DA model: classification of malignant (n = 204) vs. benign (n = 413) neoplasms (I.1); importance values of risk factors and spectral factors are plotted along different horizontal axes because of a wide value scatter. (G) gender; (A) age; (L) localization.

**Figure 2 diagnostics-12-02503-f002:**
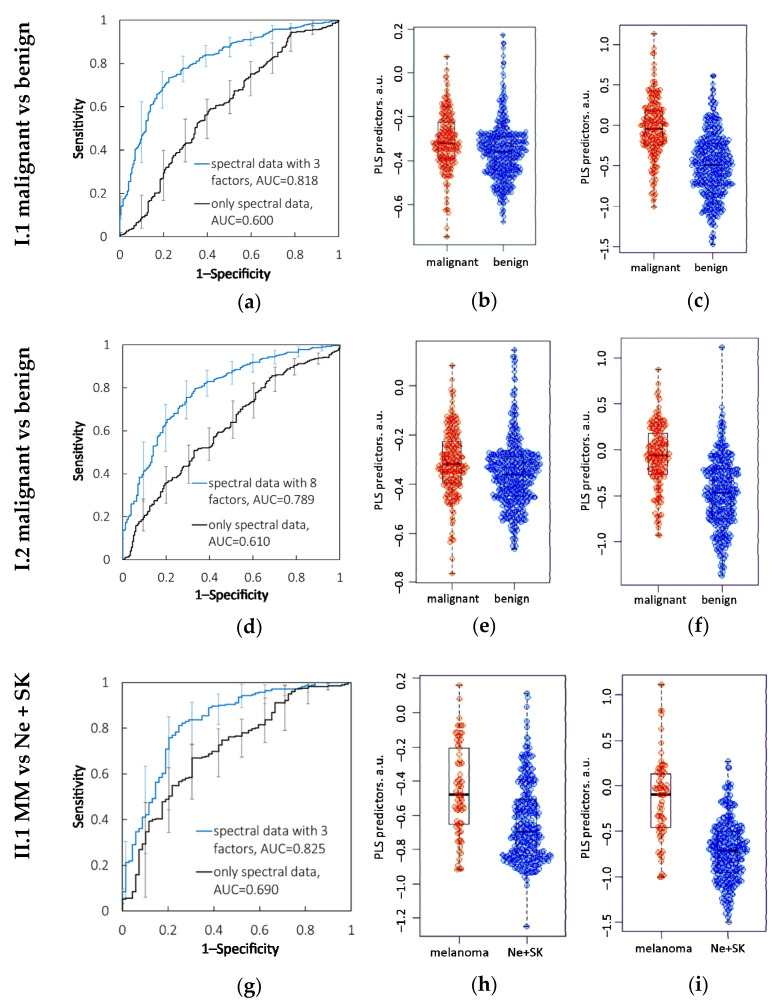
Results of diagnostic models with statistical significance (*p* < 0.05). I.1: Malignant vs. benign neoplasms: (**a**) ROC AUCs, bee-swarm diagrams of PLS predictors for tumor classification based on (**b**) spectral parameters and (**c**) combination of spectral parameters and three patient factors. II.1: Malignant vs. benign neoplasms: (**d**) ROC AUCs, bee-swarm diagrams of PLS predictors for tumor classification based on (**e**) spectral parameters and (**f**) combination of spectral parameters and eight patient factors. I.2: MM vs. benign pigmented neoplasms: (**g**) ROC AUCs, bee-swarm diagrams of PLS predictors for tumor classification based on (**h**) spectral parameters and (**i**) combination of spectral parameters and three patient factors.

**Table 1 diagnostics-12-02503-t001:** Results of regression models.

Model	ROC AUC
**I.1 Malignant (n = 204) vs. Benign (n = 413), cohort with 3 risk factors**	
only spectral data (803–914 nm)	0.600 (0.567–0.652)
spectral data with gender	0.691 (0.647–0.736), *p* = 0.008
spectral data with age	0.804 (0.767–0.840), *p* = 9 × 10^−9^
spectral data with localization	0.759 (0.718–0.800), *p* = 3 × 10^−6^
spectral data with all risk factors	0.818 (0.778–0.841), *p* = 2 × 10^−11^
**II.1 Malignant (n = 157) vs. Benign (n = 324), cohort with 8 risk factors**	
only spectral data (803–914 nm)	0.610 (0.556–0.663)
spectral data with gender	0.707 (0.658–0.756), *p* = 0.006
spectral data with age	0.718 (0.671–0.766), *p* = 0.002
spectral data with localization	0.680 (0.628–0.732), *p* = 0.035
spectral data with family history	0.625 (0.570–0.677), *p* = 0.35
spectral data with personal history	0.609 (0.556–0.663), without improvement
spectral data with sun exposure	0.609 (0.555–0.663), without improvement
spectral data with size	0.689 (0.639–0.738), *p* = 0.02
spectral data with occupational hazards	0.616 (0.563–0.669), *p* = 0.43
spectral data with all risk factors	0.789 (0.746–0.832), *p* = 5 × 10^−7^
**I.2 MM (n = 70) vs. Ne + SK (n = 283), cohort with 3 risk factors, n = 353**	
only spectral data (803–914 nm)	0.690 (0.630–0.761)
spectral data with gender	0.751 (0.685–0.818), *p* = 0.2
spectral data with age	0.771 (0.706–0.837), *p* = 0.1
spectral data with localization	0.772 (0.709–0.835), *p* = 0.1
spectral data with all risk factors	0.825 (0.766–0.884), *p* = 0.02
**II.2 MM (n = 49) vs. Ne + SK (n = 221) (cohort with 8 risk factors, n = 270)**	
only spectral data (803–914 nm)	0.789 (0.718–0.861)
spectral data with gender	0.801 (0.729–0.873), *p* = 0.4
spectral data with age	0.808 (0.734–0.881), *p* = 0.37
spectral data with localization	0.804 (0.737–0.871), *p* = 0.4
spectral data with family history	0.796 (0.726–0.866), *p* = 0.45
spectral data with personal history	0.744 (0.668–0.819), without improvement
spectral data with sun exposure	0.798 (0.725–0.870), *p* = 0.44
spectral data with size	0.806 (0.736–0.876), *p* = 0.38
spectral data with occupational hazards	0.788 (0.714–0.861), without improvement
spectral data with all risk factors	0.849 (0.785–0.914), *p* = 0.14
**I.3 MM (n = 70) vs. SK (n = 113) (cohort with 3 risk factors, n = 183)**	
only spectral data (803–914 nm)	0.791 (0.728–0.859)
spectral data with gender	0.791 (0.722–0.859), without improvement
spectral data with age	0.791 (0.723–0.859), without improvement
spectral data with localization	0.841 (0.783–0.900), *p* = 0.15
spectral data with all risk factors	0.844 (0.786–0.902), *p* = 0.15
**II.3 MM (n = 49) vs. SK (n = 90) (cohort with 8 risk factors, n = 139)**	
only spectral data (803–914 nm)	0.814 (0.740–0.888)
spectral data with gender	0.815 (0.741–0.889), *p* = 0.49
spectral data with age	0.815 (0.740–0.889), *p* = 0.49
spectral data with localization	0.851 (0.784–0.918), *p* = 0.25
spectral data with family history	0.816 (0.743–0.889), *p* = 0.48
spectral data with personal history	0.815 (0.742–0.889), *p* = 0.49
spectral data with sun exposure	0.815 (0.741–0.889), *p* = 0.49
spectral data with size	0.860 (0.795–0.925), *p* = 0.19
spectral data with occupational hazards	0.815 (0.740–0.889), *p* = 0.49
spectral data with all risk factors	0.820 (0.748–0.892), *p* = 0.46

## Data Availability

The data presented in this study are available on request from the corresponding author.

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
