# Peer review of "Combination of Optical Biopsy with Patient Data for Improvement of Skin Tumor Identification"

_diagnostics, 2022, doi:10.3390/diagnostics12102503_

Round 1

Reviewer 1 Report

Dear Authors,

1.      Line 120, do not start sentence with numerical.

2.      Text seems to more in introduction, results. It is better to be concise.

3.      It is better to provide images for better understanding of the script.

Author Response

We would like to acknowledge the Reviewers for his work and provided critical comments. We made changes in the paper according to provided reviewers’ comments. All changes in the paper are highlighted with yellow color.

Below, please find point-by-point answers to 1 Reviewer’ comments.

Comments and Suggestions for Authors (Reviewer 1)

  1. Line 120, do not start sentence with numerical.

Answer:

The corresponding remark was corrected.

  1. Text seems to more in introduction, results. It is better to be concise.

Answer:

Text in introduction has been partially shortened.

  1. It is better to provide images for better understanding of the script.

Answer:

The averaged fluorescence and Raman spectra of malignant and benign neoplasm cases included in this study were presented in our previously work [https://doi.org/10.1111/exd.14301]. We did not duplicate this figures so as not to clutter up the manuscript. Figures with patients’ statistics can be found in Supplementary files (Figure S1).

Reviewer 2 Report

None

Author Response

We would like to acknowledge the Reviewer for his work and provided critical comments. We made changes in the paper according to provided Reviewers’ comments. All changes in the paper are highlighted with yellow color.

Comments and Suggestions for Authors from Reviewer 2: None

Reviewer 3 Report

In this manuscript, the authors combine data related to patients demographics (age, gender), location of the lesion and other risk factors (family and personal history, sun exposure, occupational hazards, size of the lesion) with data retrieved from Raman and autofluorescence spectral analysis to better classify malignant versus benign neoplasms in human skin. The authors demonstrate that adding the data related to patient demographics, lesion location and/or risk factors to the spectral data significantly improve the classification of malignant versus benign neoplasms and melanoma from pigmented benign skin tumors. This approach did not significantly distinguished melanoma from seborrheic keratosis. The analysis was performed by the assessment of the area under the receiver operating characteristic (ROC) curve.

This study is novel in that it provides an in depth discussion of a broad range of patient risk factors used in conjunction with an optical spectroscopy method to significantly improve its diagnosis performance for malignant skin tumors. However, the rationale for the study and thus, its significance, is somewhat unclear. The current standard of care for clinical diagnosis of skin lesions is dermoscopy with consideration of patient demographics and risk factors (see for instance, Zalaudek I et al. Arch Dermatol. 2009;145(7):816–826). This manuscript is focused on demonstrating that patient demographics and risk factors improve the performance of the proposed spectral detection approach for the diagnosis of malignant skin lesions, including melanoma. It is assumed this proposed approach is intended as an alternative to the existing standard of care. Yet, there is no comparison with the standard of care. In other words, what is the significance of the improved spectral analysis approach? Does it perform better than the current clinical diagnosis method? This aspect needs to be clarified even if these comparison data may not be available at this early stage of the proposed approach.

Additional issues that need to be addressed:

-       The abstract does not fully summarize the work described in the manuscript, mainly it only states the results based on adding the patient demographics and lesion location when in fact, the work included additional risk factors.

-       Figure 1 is not very informative. It is unclear what the spectrum represents. The figure caption does not describe it, is it from a particular lesion? The figure should include the Raman spectra from multiple lesions, particularly malignant and benign ones for comparison. The figure caption needs to be revised to describe what is in the figure and to remove information that is not related to what the figure shows.

-       The implication of the approach not being able to distinguish melanoma from SKs needs to be briefly addressed

Author Response

We would like to acknowledge the Reviewers for his work and provided critical comments. We made changes in the paper according to provided reviewers’ comments. All changes in the paper are highlighted with yellow color.

Comments and Suggestions for Authors from Reviewer 3:

In this manuscript, the authors combine data related to patients demographics (age, gender), location of the lesion and other risk factors (family and personal history, sun exposure, occupational hazards, size of the lesion) with data retrieved from Raman and autofluorescence spectral analysis to better classify malignant versus benign neoplasms in human skin. The authors demonstrate that adding the data related to patient demographics, lesion location and/or risk factors to the spectral data significantly improve the classification of malignant versus benign neoplasms and melanoma from pigmented benign skin tumors. This approach did not significantly distinguished melanoma from seborrheic keratosis. The analysis was performed by the assessment of the area under the receiver operating characteristic (ROC) curve.

This study is novel in that it provides an in depth discussion of a broad range of patient risk factors used in conjunction with an optical spectroscopy method to significantly improve its diagnosis performance for malignant skin tumors. However, the rationale for the study and thus, its significance, is somewhat unclear. The current standard of care for clinical diagnosis of skin lesions is dermoscopy with consideration of patient demographics and risk factors (see for instance, Zalaudek I et al. Arch Dermatol. 2009;145(7):816–826). This manuscript is focused on demonstrating that patient demographics and risk factors improve the performance of the proposed spectral detection approach for the diagnosis of malignant skin lesions, including melanoma. It is assumed this proposed approach is intended as an alternative to the existing standard of care. Yet, there is no comparison with the standard of care. In other words, what is the significance of the improved spectral analysis approach? Does it perform better than the current clinical diagnosis method? This aspect needs to be clarified even if these comparison data may not be available at this early stage of the proposed approach.

Answer:

Also, it is interesting to compare the results of the proposed methodology and the results of dermoscopic images analysys performed by dermatologists, which are the current standard of care for clinical diagnosis of skin lesions. According to the Ref. [10.1016/j.annonc.2019.10.013%20] the accuracy of melanoma vs non-melanoma skin lesions classification was 79.9% for novice dermatologists, 83.3% for qualified dermatologists, and 86.9% for experts. The mean diagnostics performance of 21 board certified dermatologists using dermoscopic images to classify 71 malignant vs 40 benign lesions was nearly 71% sensitivity and 81% specificity [10.1038/nature21056]. Thus, the proposed methodology is able to classify skin neoplasms with a mean accuracy higher than the accuracy of GPs and trainees, but slightly less accuracy in comparison with trained dermatologists and experts.

The corresponding correction was added to the Discussion (382-393 Lines).

Additional issues that need to be addressed:

-       The abstract does not fully summarize the work described in the manuscript, mainly it only states the results based on adding the patient demographics and lesion location when in fact, the work included additional risk factors.

Answer:

The corresponding correction was added to the Abstract (20-22 lines).

-       Figure 1 is not very informative. It is unclear what the spectrum represents. The figure caption does not describe it, is it from a particular lesion? The figure should include the Raman spectra from multiple lesions, particularly malignant and benign ones for comparison. The figure caption needs to be revised to describe what is in the figure and to remove information that is not related to what the figure shows.

Answer:

The averaged fluorescence and Raman spectra of malignant and benign neoplasm cases included in this study were presented in our previously work [https://doi.org/10.1111/exd.14301].

The distribution of VIP scores as a weighted sum of loadings are shown in Fig.1 and solely highlights all spectral features for all loadings obtained in this PLS classification model [10.1016/j.chemolab.2004.12.011]. The VIP scores show the importance of the variables for predicting the reliability of the model. The higher the VIP score value the more important the corresponding variable is in the PLS model. In our work, the VIP scores are able to find the informative bands of the neoplasm spectra in regression specification to classify different tumor types.

Description of VIP scores is presented in Subsection 2.4 (184-188 Lines) and also briefly added in Results (204-208 Lines).

-       The implication of the approach not being able to distinguish melanoma from SKs needs to be briefly addressed.

Answer:

To improve diagnostic performance, proposed methodology may be added to the esti-mation of neoplasm morphology performed during dermoscopy analysis. Especially, deep learning-based applications using computer vision show promising results in de-tecting melanoma based on the analysis of the dermoscopic images [10.1016/j.annonc.2019.10.013, 10.1038/nature21056, 10.3390/diagnostics12092115]. However, additional studies are required to estimate capability of joint dermoscopy analysis and low-cost Raman systems.

The corresponding correction was added to the Discussion (405-410 Lines).

Reviewer 4 Report

A brief summary

The manuscript entitled ‘Combination of optical biopsy with patient data for improvement of skin tumors identification’ represents the interesting approach that would help physicians to accurate diagnose of skin tumors. However, in my opinion before publication some results should be included and a more detailed description of experimental procedure should be provided.

 General concept comments

The authors cite statistical information, articles and reviews published before 2002 (e.g. ref. 9, 13-15, 17, 21). The introduction should be significantly improved by citing the latest literature.

It is not clear whether the optical features of healthy skin located near lesion were taken into account for analysis Raman and fluorescence data. It is well known that the optical features of skin depend on Fitzpatrick skin type, location on the body, patient age and using of beauty aids. Were these aspects considered in the study?

For models I.1 and II.1, the authors combine both pigmented and non-pigmented lesions into groups, which, in my opinion, is not correct. This is confirmed by the value of ROC AUC for these models (only spectral data) that less than 0.7 and is not acceptable for discrimination. In practice physicians use different algorithms for diagnosing pigmented and non-pigmented lesions. In addition, the presence of pigment affects the optical features of lesions.

All of the lesions considered in this study were found to be suspicious of cancer. In my opinion, the study should include groups of lesions diagnosed by dermoscopy as malignant and benign (but not suspicious).

It is not very clear, how many areas on the lesion did authors use to record spectral data. In case of a single area, how was it choose, especially on large lesions?

 Specific comments

The spectral ranges indicated in the materials and methods (803-914 nm) and in the table 1 (800-914 nm) are differ. Please correct or comment.

There is some deviation in number of patient presented in the Table 1 and in the subsection "Preprocessing and Statistical Analysis of Spectra" of Material and methods:

“MM (n=70) vs benign pigmented neoplasms (Ne and SK, n=283) neoplasms 169 with 3 risk factors” in Mat&Met and “I.2 MM (n=70) vs Ne+SK (n=282), cohort with 3 risk factors, n=351” in table 1. It should be noted, that 70+282≠351.

“II.2 MM (n=49) vs benign pigmented neoplasms (Ne and SK, n=221) neoplasms 171 with 8 risk factors” in Mat&Met and “II.2 MM (n=49) vs Ne+SK (n=172) (cohort with 8 risk factors, n=221)” in table 1.

“MM (n=70) vs SK (n=113) (cohort with 3 risk factors, n=182)” in table 1. It should be noted, that 70+113≠182.

Please check and correct.

Author Response

We would like to acknowledge the Reviewer for his work and provided critical comments. We made changes in the paper according to provided Reviewer’ comments. All changes in the paper are highlighted with yellow color.

Below, please find point-by-point answers to 4 Reviewer’ comments.

Comments and Suggestions for Authors from Reviewer 4:

General concept comments.

  • The authors cite statistical information, articles and reviews published before 2002 (e.g. ref. 9, 13-15, 17, 21). The introduction should be significantly improved by citing the latest literature.

Answer:

The references were updated in the manuscript.

  • It is not clear whether the optical features of healthy skin located near lesion were taken into account for analysis Raman and fluorescence data. It is well known that the optical features of skin depend on Fitzpatrick skin type, location on the body, patient age and using of beauty aids. Were these aspects considered in the study?

Answer:

Multivariate analysis of a skin spectra of different classes automatically selects and identifies a neoplasm based on a statistically significant spectral contrast of classes. This allows us to state that the fluorescence and Raman data include all useful information on the chemical composition features of the neoplasm, including the localization features. Also, location on the body and patient age are were digitized and included as additional predictors for skin cancer identification in statistical model. Results of the significance of these factors are summarized in Table 1.

  • For models I.1 and II.1, the authors combine both pigmented and non-pigmented lesions into groups, which, in my opinion, is not correct. This is confirmed by the value of ROC AUC for these models (only spectral data) that less than 0.7 and is not acceptable for discrimination. In practice physicians use different algorithms for diagnosing pigmented and non-pigmented lesions. In addition, the presence of pigment affects the optical features of lesions.

Answer:

Optical biopsy based on Raman and autofluorescence spectroscopy may be used as universal technology to identify malignant tumors of any type as its effectiveness depends not only on pigment concentration in the tumor. It may be insists that the alteration of spectral data (especially for Raman) rely on complex biochemical composition of the tumor, which make it possible to determine as pigmented as non-pigmented lesions. For example, in our previous work [10.18287/JBPE21.07.020308], we achieved that spectral analysis in the NIR region allows one to define amelanocytic melanoma as true malignant melanoma despite the differences of pigment (melanin) presence. This finding is based on the similarity of amelanocytic melanoma and pigmented melanoma Raman spectral properties.

  • All of the lesions considered in this study were found to be suspicious of cancer. In my opinion, the study should include groups of lesions diagnosed by dermoscopy as malignant and benign (but not suspicious).

Answer:

This study included all cases of malignant and benign skin tumors that were treated by patients during the experiment in oncological center but not only suspicious. For some of them, the preliminary diagnosis established by the oncologist was confirmed by the results of histological analysis. However, to test the performance of the optical biopsy, all cases were included in this study.

  • It is not very clear, how many areas on the lesion did authors use to record spectral data. In case of a single area, how was it choose, especially on large lesions?

Answer:

Studying of the skin tumor cases was performed within the large clinical study in Samara Clinical Oncology Dispensary. In accordance with hospital capacities, the examination time by a doctor for one patient was limited to 20 minutes and during this time it is necessary to perform the spectral measurements. In results, we had time to register only one spectral measurement from tumor area. Therefore, we have not spectral data to present the spectral variations within the lesion. However, the results of the our previously work [10.1117/1.JBO.20.2.025003] have demonstrated that spectral analysis of the whole tumor is redundant for cancer type identification especially for tumor with homogenous morphological features. It is enough to check the area in the center of the tumor. Therefore, the spectral measurement of each skin tumor was registered from nearly the central tumor area point. Moreover, region of interest for Raman registration of tumor was confirmed by a medical specialist on the basis of dermatoscopic image.

The corresponding remark was added to the Subsection 2.2 (113-115 Lines).

  • Specific comments

The spectral ranges indicated in the materials and methods (803-914 nm) and in the table 1 (800-914 nm) are differ. Please correct or comment.

There is some deviation in number of patient presented in the Table 1 and in the subsection "Preprocessing and Statistical Analysis of Spectra" of Material and methods:

“MM (n=70) vs benign pigmented neoplasms (Ne and SK, n=283) neoplasms 169 with 3 risk factors” in Mat&Met and “I.2 MM (n=70) vs Ne+SK (n=282), cohort with 3 risk factors, n=351” in table 1. It should be noted, that 70+282≠351.

“II.2 MM (n=49) vs benign pigmented neoplasms (Ne and SK, n=221) neoplasms 171 with 8 risk factors” in Mat&Met and “II.2 MM (n=49) vs Ne+SK (n=172) (cohort with 8 risk factors, n=221)” in table 1.

“MM (n=70) vs SK (n=113) (cohort with 3 risk factors, n=182)” in table 1. It should be noted, that 70+113≠182.

Please check and correct.

Answer:

The corresponding remarks were corrected in the manuscript text.
